# Rural-urban outcome differences associated with COVID-19 hospitalizations in North Carolina

**Sheri Denslow**[1☯], **Jason R. Wingert**[2☯]*, **Amresh D. Hanchate**[3], **Aubri Rote**[2], **Daniel Westreich**[4], **Laura Sexton**[5], **Kedai Cheng**[6], **Janis Curtis**[7], **William Schuyler Jones**[8], **Amy Joy Lanou**[9], **Jacqueline R. Halladay**[10]

1 Epidemiologist/Statistician, Department of Research, UNC Health Sciences at MAHEC, Asheville, North Carolina, United States of America, 2 Department of Health and Wellness, University of North Carolina Asheville, Asheville, NC, United States of America, 3 Social Sciences and Health Policy, Wake Forest University School of Medicine, Winston-Salem, North Carolina, United States of America, 4 Department of Epidemiology, University of North Carolina at Chapel Hill, Chapel Hill, North Carolina, United States of America, 5 Sage Nutrition Associates, University of North Carolina Asheville, Asheville, North Carolina, United States of America, 6 Department of Mathematics, University of North Carolina Asheville, Asheville, North Carolina, United States of America, 7 Clinical Data Research Networks, Duke University, Durham, North Carolina, United States of America, 8 Associate Professor of Medicine, Associate Professor of Population Health Sciences, Member of the Duke Clinical Research Institute, Duke University, Durham, North Carolina, United States of America, 9 Department of Health and Wellness, Executive Director, NC Center for Health and Wellness, University of North Carolina Asheville, Asheville, North Carolina, United States of America, 10 Department of Family Medicine, University of North Carolina at Chapel Hill, Chapel Hill, North Carolina, United States of America

☯ These authors contributed equally to this work.
* jwingert@unca.edu

**Data Availability Statement:** Regarding the public sharing of data related to this study, UNC Chapel Hill executed Data Use Agreements (DUAs) with our contributing health systems (Duke Health and

## Abstract

People living in rural regions in the United States face more health challenges than their non-rural counterparts which could put them at additional risks during the COVID-19 pandemic. Few studies have examined if rurality is associated with additional mortality risk among those hospitalized for COVID-19. We studied a retrospective cohort of 3,991 people hospitalized with SARS-CoV-2 infections discharged between March 1 and September 30, 2020 in one of 17 hospitals in North Carolina that collaborate as a clinical data research network. Patient demographics, comorbidities, symptoms and laboratory data were examined. Logistic regression was used to evaluate associations of rurality with a composite outcome of death/hospice discharge. Comorbidities were more common in the rural patient population as were the number of comorbidities per patient. Overall, 505 patients died prior to discharge and 63 patients were discharged to hospice. Among rural patients, 16.5% died or were discharged to hospice vs. 13.3% in the urban cohort resulting in greater odds of death/hospice discharge (OR 1.3, 95% CI 1.1, 1.6). This estimate decreased minimally when adjusted for age, sex, race/ethnicity, payer, disease comorbidities, presenting oxygen levels and cytokine levels (adjusted model OR 1.2, 95% CI 1.0, 1.5). This analysis demonstrated a higher COVID-19 mortality risk among rural residents of NC. Implementing policy changes may mitigate such disparities going forward.

Wake Forest University) to provide for use of these data. These DUAs prohibit us from sharing the data outside of UNC Chapel Hill. Additionally, the University of North Carolina at Chapel Hill also employs strict data security controls which require that the data be stored and analyzed on a UNC Chapel Hill provided secure research workspace. No investigators have the authority to remove these data from the secure workspace for sharing or any purpose. While we understand and appreciate the journal's data sharing policy, we are unable to provide these data to the journal due to these legal and security restrictions. A letter from Mr. Andy Johns has been uploaded along with this submission to detail our legal restriction to the public sharing of data. The University of North Carolina at Chapel Hill can be listed as the institutional body to which general data requests can be sent and emails communications can go to andy_johns@unc.edu.

**Funding:** The North Carolina Policy Collaboratory, awarded to Dr. Amy Joy Lanou. The North Carolina Collaboratory award does not have an award number, but the grant's title is Title Back-to-College Challenge: Health Ambassadors for a Coordinated Culture of Safety and Wellness on WNC Campuses and Statewide Co-Morbidity Study with funding from 7/1/2020-12/30/2020. The project described was also supported by the National Center for Advancing Translational Sciences (NCATS), National Institutes of Health, through Grant Award Number UL1TR002489. The content is solely the responsibility of the authors and does not necessarily represent the official views of the NIH. The funders had no role in study design, data collection and analysis, decision to publish, or preparation of the manuscript.

**Competing interests:** The authors have declared that no competing interests exist.

## Introduction

Coronavirus Disease 2019 (COVID-19), caused by the novel severe acute respiratory syndrome coronavirus 2 (SARS-CoV-2), continues to be a global pandemic. As of July 22, 2021, COVID-19 has resulted in 607,289 deaths in the US. Although numerous analyses have been devoted to understanding what patient comorbidities, demographics, behaviors, laboratory values and medical interventions received are associated with dying of COVID-19 [1–5], few studies have specifically tried to determine if "rurality" of a patient's residence increases mortality risk [6–9] and none to date have focused on residents of North Carolina (NC).

One in seven Americans reside in one of the 1,976 counties designated as rural by the 2013 National Center for Health Statistics Urban-Rural Classification Scheme for Counties in 2018 [10]. Rural populations in the US have shorter life expectancies [11], lower median incomes [11], greater prevalence of comorbid health conditions such as cancer, heart disease, diabetes, hypertension, and obesity [12, 13]. They are also older [14] with a population mean age of 51 in rural compared to 45 in urban regions [15]. As well, they have lower access to healthcare: only 1% of the nation's ICU beds are located in rural areas [16]. Over 4.7 million people live in 460 rural counties across the nation where there are no general medical or surgical hospital beds, and 16.4 million people live in rural areas with no medical/surgical intensive care unit (ICU) beds [17]. Health care facilities within rural communities are typically less resourced with reduced access to personal protective equipment, ICU beds, testing, and the necessary equipment to effectively treat people most severely affected by COVID-19 infection complications, which are commonly older adults [18]. As a result, many rural hospitals find themselves needing to transfer residents with more serious cases of COVID-19 to larger facilities in urban areas for treatment [19]. Hospital transfers require time, and that can affect disease outcomes in critical situations. Relocating patients to urban areas may present additional challenges if the receiving hospital is already overwhelmed [20].

With rural communities at a notable disadvantage in terms of COVID-19 health outcomes related to healthcare and population demographics, and with COVID-19 proving to be a more intense burden on older populations, we hypothesize that individuals in rural areas will face more risk of COVID-19-associated death compared to comparable individuals in urban areas [7, 21].

NC has 21.2% of its people living in rural areas [22]. Similar to national trends, NC rural areas are inhabited by people who are older, and more likely to be uninsured [23]. NC reported a significant increase in COVID-19 burden in rural areas in September 2020, with rural areas making up the majority of state cases and deaths [24]. NC has also endured 7 rural hospital closures since 2010 [25]. Thus, it is important to understand NC-level data specific to rurality.

The purpose of this study is to understand if rural patients with COVID-19 experienced a different risk of death compared to their urban counterparts. This study describes the demographics, baseline comorbidities, clinical test results, and deaths among hospitalized patients with COVID-19 in three academic health systems in NC.

## Methods

### Study setting and population

We identified hospital patients with infection and/or a COVID-19 diagnosis who received care in one of three large, NC-based academic health systems: University of North Carolina Chapel Hill (UNC Chapel Hill), Duke University and Wake Forest Baptist Health. These three health systems have 17 hospitals spanning 13 counties of NC (Fig 1) and are part of a distributed research network named PCORnet, the National Patient-Centered Clinical Research

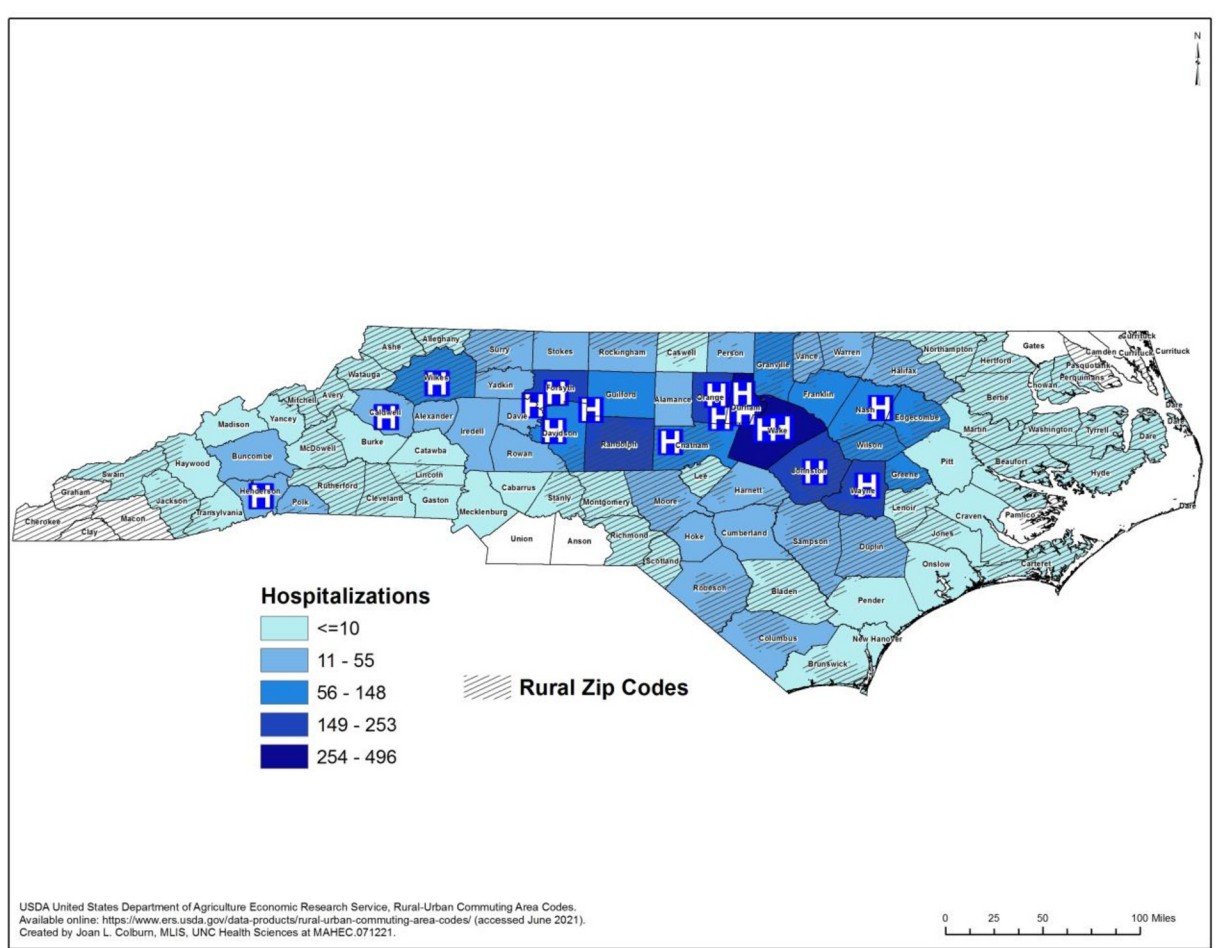

**Fig 1. Map of hospitalizations by rural and urban zip codes.** Map of rural and urban zip codes with the number of COVID-19 hospitalizations per county between March 1 and September 30, 2020. Locations of hospitals included in this study are shown with an "H".

Network® [26]. PCORnet is funded in part by the Patient Centered Outcomes Research Institute where institution-specific patient data such as vital signs, demographics, laboratory test results and care delivered are harmonized. In NC, many larger health systems have invested in methods to harmonize and aggregate hospital level data generated initially from electronic health record data in order to better understand the care processes and outcomes of patients. UNC Health, Duke and Wake Forest Baptist Health have worked for over 10 years to engage in such efforts to support multi-site research studies and during the COVID-19 pandemic, developed limited data sets, as defined by the Health Insurance Portability and Accountability Act (HIPAA), to better understand outcomes of patients cared for in consortium hospitals. As a result, those data have a common format to facilitate data aggregation, multi-site involvement and analyses. PCORnet uses a common data model to facilitate queries of standardized data [26] and during the COVID-19 pandemic, supported the creation of a COVID-19 specific common data model to allow for ready access to harmonized COVID-19 specific clinical data.

We collaborated with investigators at UNC-Chapel Hill to develop a COVID-19 case definition and query program, which was initially run at the UNC site and then at the Duke University and Wake Forest Baptist Health sites to identify inpatients (including emergency to inpatient stays) who had an identified SARS-CoV-2 infection and/or a COVID-19 diagnosis

and had a discharge date (which includes those that died) between March 1, 2020 and September 30, 2020. COVID-19 diagnosis was determined by the patient having a diagnosis code of B97.29, B97.21 (before April 1, 2020) or U07.1 [27]. SARS-CoV-2 identification was based on having a positive or detected status on a SARS-CoV-2 lab. For patients with multiple visits, the first inpatient encounter during the study time period with noted COVID-19 was included. People who were identified as prisoners were removed from the dataset. This study was deemed exempt from further review by the UNC-Chapel Hill, Duke University Health System, Wake Forest School of Medicine School of Medicine and UNC Asheville Institutional Review Boards. Since this study analyzed existing clinical data using a limited data set, the reviewing Institutional Review Boards deemed the data as secondary research for which consent is not required. Therefore, individual participant consent was waived for this study.

## Data elements and analysis

We electronically collected information on patient demographics (age, sex, race/ethnicity, insurance status and zip code), hospital stay characteristics (dates of admission and discharge, intensive care unit stay, ventilator use, and discharge status), smoking status, vital measurements and specific laboratory values (as listed in the tables), and all encounter-related ICD-10-CM diagnosis codes (discharge and final). The main outcome of interest was a composite of in-hospital death or discharge to hospice.

Rural status was determined from patient zip code using Rural Urban Commuting Area Codes [28] with zip codes in metropolitan areas (codes 1–3) categorized as urban and the remaining zip codes (codes 4–10) as rural. Patient zip code was also used to assign a patient county of residence using a method developed by the United States Department of Housing and Urban Development [29]. Race and ethnicity were combined into a race/ethnicity variable following the approach adopted by the Agency for Healthcare Research and Quality [30, 31]. For insurance status, patients with unknown payor who were 65 years or older were considered to have Medicare.

The Comorbidity package in R [32] was used to identify the presence of relevant comorbidities from ICD-10 diagnosis codes (acute myocardial infarction, congestive heart failure, hypertension, chronic obstructive pulmonary disease (COPD), diabetes, renal disease, cancer, liver disease, coagulopathy, or obesity) and to calculate both the Charlson [33, 34] and Elixhauser comorbidity [35] scores. The coding algorithm used by the Comorbidity package can be found in Quan et al. 2005 [36].

We conducted descriptive analyses including: percentage of patient population with assessed characteristics that had the composite outcome of death/discharge to hospice, the percentage point difference in risk of outcome from a chosen reference stratum (subtraction of percentages) and the relative risk of outcome compared to a reference stratum (division of proportions). Additionally, we used logistic regression to evaluate rurality in association with the composite outcome of odds of death/hospice discharge. To explore how much of the increase in outcome of death/hospice discharge seen for rural-dwelling individuals could be explained by socio-demographic characteristics, comorbid conditions, and presenting health status, we used a multivariable model including rural zip code (yes/no); age (age, $age^2$, $age^3$), sex (female/ male), race/ethnicity (non-Hispanic Black; Hispanic; non-Hispanic White; Other, non-Hispanic; and Missing race, non-Hispanic), insurance status (Commercial, Medicaid, Medicare, Self-pay, and missing), smoking status (current, former, never and missing), comorbidities (Charlson comorbidity index: 0, 1–2, 3–4, $> = 5$, missing), first recorded oxygen saturation with hospital or emergency room visit ($<93\%$, $> = 93\%$, missing), indicators of an inflammatory response [yes: low lymphocyte ($< = 0.8$ 10^3/uL), or elevated levels of troponin ($> = 0.1$

ng/ml), procalcitonin (>0.5 ng/mL), or C-reactive protein (CRP) (> 15 mg/dL); no: having a normal result present for any of the above listed criteria; missing: missing all 4 values] and hospital system (Duke, UNC, Wake). Analyses were performed using R version 3.5.2 (R Project for Statistical Computing; R Foundation) and SAS version 9.4 (SAS Institute Inc., Cary, NC). The map was created using ArcGIS version 10.8.1 (ESRI, Redlands, CA).

## Results

There were 3,991 inpatients with SARS-CoV-2 infection or COVID-19 diagnosis hospitalized in one of the 17 hospitals during the timeframe, with 1,977 seen at a UNC health system hospital, 1,220 at a Duke system hospital, and 735 at a Wake Forest system hospital. The majority of patients (3,856) lived in one of 89 NC counties (Fig 1). Sixty-seven patients lived out-of-state, and 68 patients were missing zip code information. Most of the included patients (76%) came from urban settings (See Table 1 for overall descriptive data including Urban vs. Rural comparisons). Hypertension was the most common comorbidity (72.2% of the study population) followed by diabetes (52.3%). The median age in the rural cohort was 63 (IQR 49–73) years and 62 (IQR 26–74) in the urban group. While Medicare was the most common insurance type overall (49% overall), Medicaid coverage was the second most common coverage for urban patients, with commercial coverage second for rural patients. Smoking status was similar across the urban and rural patient populations. All comorbidities were more common in the rural patient population as compared to the urban patient population, and the number of comorbidities per patient was also more common among the rural population which resulted in higher Charlson and Elixhauser comorbidity indices (Fig 2).

Table 2 shows the percentage of patients overall who died/were discharged to hospice within each stratum (row) by study characteristic. Overall, 505 patients (13%; rural, urban and missing zip code) died prior to discharge and 63 patients (2%) were discharged to hospice. Among patients living in rural areas, 16.5% died or were discharged to hospice vs. 13.3% in the urban cohort resulting in a 3.2 (CI 0.6–5.9) percentage point increase in death for the rural cohort. The percentage of patients who died/were discharged to hospice was higher among males and increased with age. While 3% of patients aged 35–45 died/were discharged to hospice, this was the outcome for 32% of those aged 75+. Patients with comorbidities had a higher percentage of death/hospice discharge as compared to patients without the comorbidity for all assessed conditions except for uncomplicated hypertension and obesity (Table 2). Patients

**Table 1. Demographics, characteristics and comorbidities of patients hospitalized with a SARS-CoV-2 infection or COVID-19 diagnosis, total and stratified by rural/urban zip codes.**

| | Total n = 3991 | | Urban n = 2978 | | Rural n = 945 | |
|---|---|---|---|---|---|---|
| | n | Column % | n | Column % | n | Column % |
| Age | | | | | | |
| Median age (IQR) | 62 (47–74) | | 62 (46–74) | | 63 (49–73) | |
| 0 to 17 | 114 | 2.9 | 81 | 2.7 | 33 | 3.5 |
| 18 to 34 | 426 | 10.7 | 326 | 10.9 | 90 | 9.5 |
| 35 to 44 | 370 | 9.3 | 297 | 10.0 | 71 | 7.5 |
| 45 to 54 | 520 | 13.0 | 392 | 13.2 | 116 | 12.3 |
| 55 to 64 | 795 | 19.9 | 572 | 19.2 | 207 | 21.9 |
| 65 to 74 | 843 | 21.1 | 596 | 20.0 | 234 | 24.8 |
| 75+ | 922 | 23.1 | 713 | 23.9 | 194 | 20.5 |
| Missing | 1 | 0 | 1 | 0 | 0 | 0 |
| Sex | | | | | | |

*(Continued)*

**Table 1.** (Continued)

| | Total n = 3991 | | Urban n = 2978 | | Rural n = 945 | |
|---|---|---|---|---|---|---|
| | **n** | **Column %** | **n** | **Column %** | **n** | **Column %** |
| Female | 2037 | 51.0 | 1587 | 53.3 | 432 | 45.7 |
| Male | 1953 | 48.9 | 1390 | 46.7 | 513 | 54.3 |
| Missing | 1 | 0 | 1 | 0 | 0 | 0 |
| Race/Ethnicity | | | | | | |
| Black, NH | 1365 | 34.2 | 1053 | 35.4 | 293 | 31.0 |
| Hispanic | 903 | 22.6 | 650 | 21.8 | 233 | 24.7 |
| White, NH | 1489 | 37.3 | 1095 | 36.8 | 371 | 39.3 |
| Other, NH | 166 | 4.2 | 138 | 4.6 | 28 | 3.0 |
| Unknown, NH | 68 | 1.7 | 42 | 1.4 | 20 | 2.1 |
| Insurance Status | | | | | | |
| Commercial | 492 | 12.3 | 344 | 11.6 | 148 | 15.7 |
| Medicaid | 595 | 14.9 | 458 | 15.4 | 131 | 13.9 |
| Medicare | 1962 | 49.2 | 1457 | 48.9 | 479 | 50.7 |
| Self-pay | 192 | 4.8 | 157 | 5.3 | 32 | 3.4 |
| Missing | 750 | 18.8 | 562 | 18.9 | 155 | 16.4 |
| Smoking | | | | | | |
| Current Smoker | 231 | 5.8 | 170 | 5.7 | 58 | 6.1 |
| Former smoker | 1264 | 31.7 | 949 | 31.9 | 294 | 31.1 |
| Never smoker | 2203 | 55.2 | 1680 | 56.4 | 495 | 52.4 |
| Missing | 293 | 7.3 | 179 | 6.0 | 98 | 10.4 |
| Comorbidities: | n = 3965 | | n = 2960 | | n = 938 | |
| Acute myocardial infarction | 425 | 10.7 | 302 | 10.2 | 123 | 13.1 |
| Congestive heart failure | 782 | 19.7 | 576 | 19.5 | 197 | 21.0 |
| Hypertension, uncomplicated | 1698 | 42.8 | 1240 | 41.9 | 440 | 46.9 |
| Hypertension, complicated | 1167 | 29.4 | 854 | 28.9 | 296 | 31.6 |
| COPD | 924 | 23.3 | 685 | 23.1 | 225 | 24.0 |
| Diabetes without complications | 1304 | 32.9 | 953 | 32.2 | 339 | 36.1 |
| Diabetes with complications | 770 | 19.4 | 559 | 18.9 | 203 | 21.6 |
| Renal disease | 916 | 23.1 | 666 | 22.5 | 234 | 24.9 |
| Cancer (any malignancy) | 237 | 6.0 | 171 | 5.8 | 63 | 6.7 |
| Liver disease | 241 | 6.1 | 171 | 5.8 | 64 | 6.8 |
| Coagulopathy | 702 | 17.7 | 506 | 17.1 | 180 | 19.2 |
| Obesity | 1138 | 28.7 | 847 | 28.6 | 276 | 29.4 |
| Charlson Comorbidity Score | | | | | | |
| Median (IQR) | 1 (0–3) | | 1 (0–2) | | 1 (1–3) | |
| 0 | 1090 | 27.5 | 842 | 28.4 | 230 | 24.5 |
| 1–2 | 1874 | 47.3 | 1391 | 47.0 | 442 | 47.1 |
| 3–4 | 822 | 20.7 | 596 | 20.1 | 219 | 23.3 |
| > = 5 | 179 | 4.5 | 131 | 4.4 | 47 | 5.0 |
| Elixhauser Score | | | | | | |
| Median (IQR) | 4 (2–6) | | 4 (2–6) | | 4 (2–6) | |
| 0 | 308 | 7.8 | 225 | 7.6 | 77 | 8.2 |
| 1–4 | 2027 | 51.1 | 1563 | 52.8 | 429 | 45.7 |
| > = 5 | 1630 | 41.1 | 1172 | 39.6 | 432 | 46.1 |

IQR = interquartile range; NH = non-Hispanic; COPD = chronic obstructive pulmonary disease

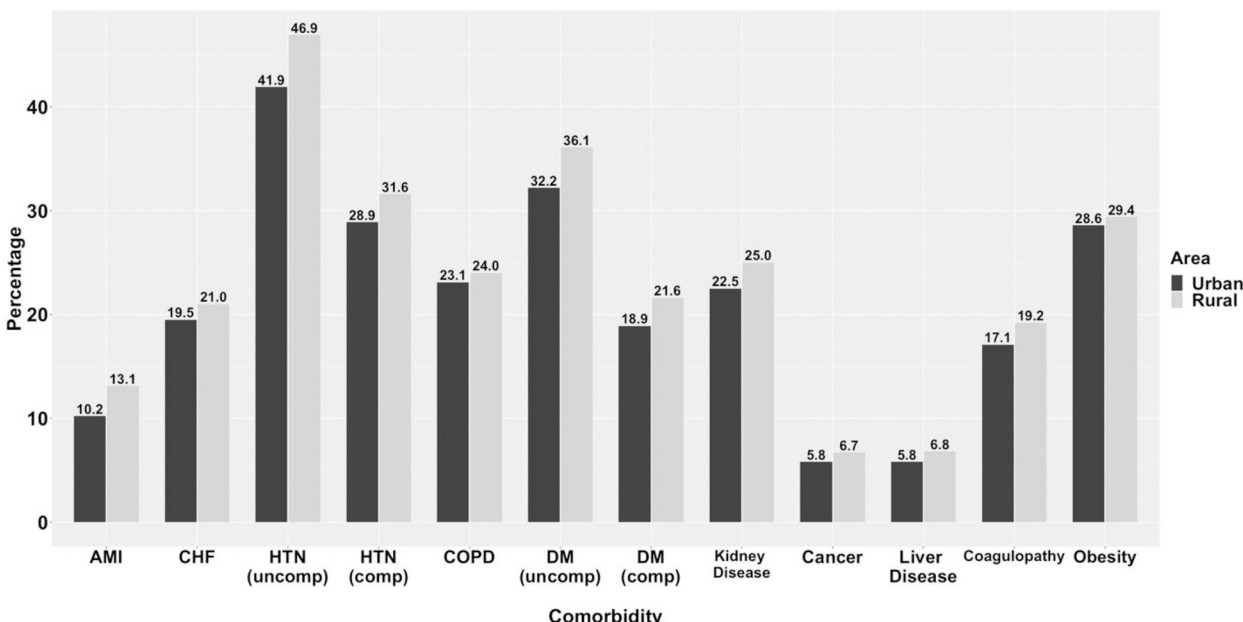

**Fig 2. Comparison of rural and urban areas by comorbidity.** Comparison of unadjusted comorbidity incidence between rural and urban residents hospitalized in North Carolina for COVID-19. AMI: Acute Myocardial Infarction, CHF: Congestive Heart Failure, HTN: Hypertension, COPD: Chronic Obstructive Pulmonary Disease, DM: Diabetes Mellitus.

**Table 2. Demographics, characteristics and comorbidities and percentage of death/hospice discharge for patients hospitalized with a SARS-CoV-2 infection or COVID-19 diagnosis.**

| | n | n death/hospice | stratum (row) % | percentage point difference (95% CI) | relative ratio (95% CI) |
|---|---|---|---|---|---|
| Rural* | | | | | |
| yes | 945 | 156 | 16.5% | 3.2 (0.6, 5.9) | 1.2 (1.0, 1.5) |
| no | 2978 | 395 | 13.3% | 0 (ref) | 1 (ref) |
| Age category | | | | | |
| 0 to 17 | 114 | <10 | ns | – | – |
| 18 to 34 | 426 | <10 | ns | – | – |
| 35 to 44 | 370 | 11 | 3.0% | 0 (ref) | 1 (ref) |
| 45 to 54 | 520 | 33 | 6.3% | 3.4 (0.7, 6.1) | 2.1 (1.1, 4.2) |
| 55 to 64 | 795 | 79 | 9.9% | 7.0 (4.3, 9.7) | 3.3 (1.8, 6.2) |
| 65 to 74 | 843 | 151 | 17.9% | 14.9 (11.8, 18.1) | 6.0 (3.3, 11.0) |
| 75+ | 922 | 290 | 31.5% | 28.5 (25.0, 31.9) | 10.6 (5.9, 19.1) |
| Sex | | | | | |
| Female | 2037 | 250 | 12.3% | 0 (ref) | 1 (ref) |
| Male | 1953 | 318 | 16.3% | 4.0 (1.8, 6.2) | 1.3 (1.1, 1.5) |
| Race/Ethnicity | | | | | |
| Black, NH | 1365 | 193 | 14.1% | -4.1 (-6.8, -1.4) | 0.8 (0.7, 0.9) |
| Hispanic | 903 | 63 | 7.0% | -11.3 (-13.9, -8.7) | 0.4 (0.3, 0.5) |
| White, NH | 1489 | 272 | 18.3% | 0 (ref) | 1 (ref) |
| Other, NH | 166 | 22 | 13.3% | -5.0 (-10.5, 0.5) | 0.7 (0.5, 1.1) |
| Unknown, NH | 68 | 18 | 26.5% | 8.2 (-2.5, 18.9) | 1.4 (1.0, 2.2) |
| Insurance Status | | | | | |
| Commercial | 492 | 27 | 5.5% | 0 (ref) | 1 (ref) |

*(Continued)*

**Table 2.** (Continued)

| | n | n death/hospice | stratum (row) % | percentage point difference (95% CI) | relative ratio (95% CI) |
|---|---|---|---|---|---|
| Medicaid | 595 | 32 | 5.4% | -0.1 (-2.8, 2.6) | 1.0 (0.6, 1.6) |
| Medicare | 1962 | 459 | 23.4% | 17.9 (15.2, 20.7) | 4.3 (2.9, 6.2) |
| Self-pay | 192 | <10 | ns | – | – |
| Smoking | | | | | |
| Current smoker | 231 | 26 | 11.3% | -0.5 (-4.8, 3.8) | 1.0 (0.7, 1.4) |
| Former smoker | 1264 | 208 | 16.5% | 4.7 (2.3, 7.2) | 1.4 (1.2, 1.7) |
| Never smoker | 2203 | 258 | 11.7% | 0 (ref) | 1 (ref) |
| Comorbidities: | | | | | |
| Acute myocardial infarction | 425 | 124 | 29.2% | 16.7 (12.2, 21.2) | 2.3 (2.0, 2.8) |
| Congestive heart failure | 782 | 198 | 25.3% | 13.8 (10.5, 17.0) | 2.2 (1.9, 2.6) |
| Hypertension, uncomplicated | 1698 | 243 | 14.3% | 0 (-2.1, 2.3) | 1.0 (0.9, 1.2) |
| Hypertension, complicated | 1167 | 273 | 23.4% | 12.9 (10.2, 15.6) | 2.2 (1.9, 2.6) |
| COPD | 924 | 164 | 17.7% | 4.5 (1.8, 7.3) | 1.3 (1.1, 1.6) |
| Diabetes without complications | 1304 | 215 | 16.5% | 3.3 (0.9, 5.7) | 1.3 (1.1, 1.5) |
| Diabetes with complications | 770 | 167 | 21.7% | 9.2 (6.1, 12.3) | 1.7 (1.5, 2.0) |
| Renal disease | 916 | 215 | 23.5% | 12.0 (9.0, 14.9) | 2.0 (1.7, 2.4) |
| Cancer (any malignancy) | 237 | 61 | 25.7% | 12.2 (6.5, 17.9) | 1.9 (1.5, 2.4) |
| Liver disease | 241 | 61 | 25.3% | 11.8 (6.2, 17.4) | 1.9 (1.5, 2.4) |
| Coagulopathy | 702 | 199 | 28.3% | 17.1 (13.6, 20.6) | 2.5 (2.2, 2.9) |
| Obesity | 1138 | 154 | 13.5% | -1.0 (-3.4, 1.3) | 0.9 (0.8, 1.1) |
| Charlson Comorbidity Score | | | | | |
| 0 | 1090 | 42 | 3.9% | 0 (ref) | 1 (ref) |
| 1–2 | 1874 | 264 | 14.1% | 10.2 (8.3, 12.2) | 3.7 (2.7, 5.0) |
| 3–4 | 822 | 193 | 23.5% | 19.6 (16.5, 22.7) | 6.1 (4.4, 8.4) |
| >= 5 | 179 | 67 | 37.4% | 33.6 (26.4, 40.8) | 9.7 (6.8, 13.8) |
| Elixhauser Score | | | | | |
| 0 | 308 | <10 | ns | – | – |
| 1–4 | 2027 | 145 | 7.2% | 0 (ref) | 1 (ref) |
| >= 5 | 1630 | 420 | 25.8% | 18.6 (16.2, 21.0) | 3.6 (3.0, 4.3) |

CI = confidence interval; ns = not shown; NH = non-Hispanic; COPD = chronic obstructive pulmonary disease

*Zip code was missing for 68 patients; patient number does not add up to 3991 and death/hospice does not add up to 505

with acute myocardial infarction and coagulopathy had the highest percentage of death/hospice discharge, each at 29%. Further, patient groups with higher Charlson and Elixhauser comorbidity scores had higher percentages of death/hospice discharge.

The majority of patients presented without fever (<100.4F), and 22% of patients had initial oxygen saturations below 93% (Table 3). Low oxygen saturation was more common in the rural patient population as compared to the urban patient population (24% versus 21%). Median BMI was 30 and was similar across both urban and rural populations. Rural patients more commonly had lower lymphocyte counts and more commonly showed signs of hyperinflammation via markers including C-reactive protein, D-dimer, and Procalcitonin.

The percentage of patients who died/were discharged to hospice was higher in those with lower initial oxygen saturation values (Table 4). Death/discharge to hospice was less common for patients with increasingly higher BMI. Patient groups with lower lymphocyte counts had higher percentages of death/hospice discharge. For markers of a hyperinflammatory response,

**Table 3. Initial vitals and labs of patients hospitalized with a SARS-CoV-2 infection or COVID-19 diagnosis, total and stratified by rural/urban zip codes.**

| | Total | | Urban | | Rural | |
|---|---|---|---|---|---|---|
| | **n** | **Column %** | **n** | **Column %** | **n** | **Column %** |
| Temperature, ˚F | n = 3792 | | n = 2819 | | n = 912 | |
| Median (IQR) | 99 (98–100) | | 99 (98–100) | | 99 (98–100) | |
| < 100.4 | 3064 | 80.8 | 2239 | 79.4 | 773 | 84.8 |
| 100.4–102.2 | 491 | 12.9 | 389 | 13.8 | 96 | 10.5 |
| >102.2 | 237 | 6.3 | 191 | 6.8 | 43 | 4.7 |
| Oxygen saturation, % | n = 3978 | | n = 2967 | | n = 943 | |
| Median (IQR) | 96 (93–98) | | 96 (93–98) | | 96 (93–98) | |
| 93–100 | 3114 | 78.3 | 2338 | 78.8 | 720 | 76.4 |
| 89–92 | 503 | 12.6 | 371 | 12.5 | 126 | 13.4 |
| < = 88 | 361 | 9.1 | 258 | 8.7 | 97 | 10.3 |
| BMI | n = 3149 | | n = 2368 | | n = 730 | |
| Median (IQR) | 30 (25–36) | | 30 (25–36) | | 30 (25–36) | |
| <18.5 | 100 | 3.2 | 68 | 2.9 | 30 | 4.1 |
| 18.5–24.9 | 631 | 20.0 | 482 | 20.4 | 136 | 18.6 |
| 25–29.9 | 827 | 26.3 | 620 | 26.2 | 191 | 26.2 |
| 30–39.9 | 1150 | 36.5 | 858 | 36.2 | 277 | 37.9 |
| 40+ | 441 | 14.0 | 340 | 14.4 | 96 | 13.2 |
| Lymphocyte, ($10^3$/ul) | n = 3600 | | n = 2699 | | n = 838 | |
| Median (IQR) | 0.9 (0.6–1.4) | | 1.0 (0.7–1.4) | | 0.9 (0.6–1.3) | |
| >1.2 | 1100 | 30.6 | 846 | 31.3 | 236 | 28.2 |
| >0.8–1.2 | 989 | 27.5 | 768 | 28.5 | 206 | 24.6 |
| 0.5–0.8 | 1113 | 30.9 | 805 | 29.8 | 287 | 34.2 |
| <0.5 | 398 | 11.1 | 280 | 10.4 | 109 | 13.0 |
| Aspartate aminotransferase, U/L | n = 3665 | | n = 2732 | | n = 868 | |
| Median (IQR) | 37 (26–58) | | 37 (26–57) | | 39 (27–61) | |
| < = 33 | 1530 | 41.7 | 1150 | 42.1 | 356 | 41.0 |
| >33 | 2135 | 58.3 | 1582 | 57.9 | 512 | 59.0 |
| Alanine aminotransferase, U/L | n = 3667 | | n = 2735 | | n = 868 | |
| Median (IQR) | 27 (18–45) | | 27 (18–45) | | 27 (18–46) | |
| < = 34 | 2318 | 63.2 | 1730 | 63.3 | 548 | 63.1 |
| >34 | 1349 | 36.8 | 1005 | 36.7 | 320 | 36.9 |
| Creatinine, mg/dL | n = 3800 | | n = 2837 | | n = 895 | |
| Median (IQR) | 1.0 (0.8–1.5) | | 1.0 (0.8–1.4) | | 1.0 (0.8–1.6) | |
| 0–1.1 | 2313 | 60.9 | 1761 | 62.1 | 517 | 57.8 |
| > 1.1–2 | 912 | 24.0 | 674 | 23.8 | 221 | 24.7 |
| >2 | 575 | 15.1 | 402 | 14.2 | 157 | 17.5 |
| Troponin, ng/mL | n = 2248 | | n = 1675 | | n = 549 | |
| Median (IQR) | 0.03 (0.02–0.10) | | 0.03 (0.01–0.11) | | 0.03 (0.02–0.09) | |
| <0.1 | 1681 | 74.8 | 1244 | 74.3 | 416 | 75.8 |
| 0.1–1 | 507 | 22.6 | 390 | 23.3 | 114 | 20.8 |
| >1 | 60 | 2.7 | 41 | 2.5 | 19 | 3.5 |
| Procalcitonin, ng/mL | n = 1619 | | n = 1313 | | n = 292 | |
| Median (IQR) | 0.2 (0.1–0.4) | | 0.2 (0.1–0.4) | | 0.2 (0.1–0.5) | |
| 0–0.5 | 1267 | 78.3 | 1035 | 78.8 | 223 | 76.4 |
| >0.5 | 352 | 21.7 | 278 | 21.2 | 69 | 23.6 |
| D-Dimer, ng/mL | n = 514 | | n = 285 | | n = 229 | |

*(Continued)*

**Table 3.** (Continued)

| | Total | | Urban | | Rural | |
|---|---|---|---|---|---|---|
| | **n** | **Column %** | **n** | **Column %** | **n** | **Column %** |
| median (IQR) | 474 (249–1066) | | 422 (236–955) | | 540 (264–1194) | |
| < = 500 | 265 | 51.6 | 159 | 55.8 | 106 | 46.3 |
| 501–1000 | 111 | 21.6 | 56 | 19.6 | 55 | 24.0 |
| >1000 | 138 | 26.8 | 70 | 24.6 | 68 | 29.7 |
| Ferritin, ng/mL | n = 2329 | | n = 1687 | | n = 598 | |
| Median (IQR) | 439 (200–877) | | 434 (199–869) | | 462 (199–900) | |
| 0–250 | 713 | 30.6 | 518 | 30.7 | 182 | 30.4 |
| >250–500 | 567 | 24.3 | 419 | 24.8 | 134 | 22.4 |
| >500–1000 | 545 | 23.4 | 383 | 22.7 | 152 | 25.4 |
| >1000–2500 | 382 | 16.4 | 281 | 16.7 | 96 | 16.1 |
| >2500 | 122 | 5.2 | 86 | 5.1 | 34 | 5.7 |
| C-reactive protein, mg/L | n = 2227 | | n = 1624 | | n = 564 | |
| Median (IQR) | 71 (24–155) | | 68 (21–148) | | 79 (32–172) | |
| 0–15 | 411 | 18.5 | 324 | 20.0 | 81 | 14.4 |
| >15–100 | 949 | 42.6 | 697 | 42.9 | 242 | 42.9 |
| >100–200 | 533 | 23.9 | 378 | 23.3 | 139 | 24.6 |
| >200 | 334 | 15 | 225 | 13.9 | 102 | 18.1 |

IQR = interquartile range; BMI = body mass index

except for alanine aminotransferase, patient groups with higher values had a larger percentage of death/discharge to hospice.

Further evaluation of rurality with the composite outcome of odds of death/hospice discharge using logistic regression demonstrated that rural patients had 1.3 times the odds of dying/hospice discharge as compared to urban patients (crude model: 95% CI 1.1, 1.6) The estimate decreased minimally when adjusted for age, sex, race/ethnicity, payer, disease comorbidities, presenting oxygen levels and cytokine levels, and hospital system (adjusted model OR 1.2, 95% CI 1.0, 1.5).

## Discussion

This study described the characteristics of adults hospitalized with COVID-19 (identified through laboratory-confirmed SARS-CoV-2 infection and/or a COVID-19 diagnosis) in three large healthcare systems (17 total hospitals) in North Carolina from the time of COVID-19 pandemic onset through September, 2020. Of the 3,991 patients included in this study during that time period, 13% died and 2% were admitted to hospice care. The overall mortality rate observed here was lower than that observed in other large populations [5], despite higher prevalence of most measured comorbidities in our population. Notably, these hospital-level data revealed several urban-rural differences. Urban hospitalized patients were more likely to be female, with the opposite true in rural areas. Patients from urban areas were more often insured by Medicaid, while those from rural areas were more often insured by commercial insurers or Medicare. However, ages were similar for rural and urban patients.

Most importantly, patients from rural areas were slightly more likely to die from COVID-19 or be discharged to hospice than those from urban areas (death/hospice discharge percentage: 16.5% in rural vs 13.3% in urban). We found that even after adjustment for individual characteristics, such as age, sex, race, ethnicity, insurance provider, smoking status,

**Table 4. Initial vitals and labs and percentage of death/hospice discharge for patients hospitalized with a SARS-CoV-2 infection or COVID-19 diagnosis.**

| | N | N death/hospice | stratum % | bar graph of stratum % | % point difference | RR |
|---|---|---|---|---|---|---|
| **Temperature ˚F** | | | | | | |
| < 100.4˚F (<38˚C) | 3064 | 430 | 14.0 | 14.0 | 0 (ref) | 1 (ref) |
| 100.4˚F-102.2˚F (38–39˚C) | 491 | 73 | 14.9 | 14.9 | 0.8 (-2.5, 4.2) | 1.1 (0.8, 1.3) |
| >102.2˚F (>39˚C) | 237 | 41 | 17.3 | 17.3 | 3.3 (-1.7, 8.2) | 1.2 (0.9, 1.7) |
| **Oxygen saturation** | | | | | | |
| 93–100% | 3114 | 366 | 11.8 | 11.8 | 0 (ref) | 1 (ref) |
| 89–92% | 503 | 89 | 17.7 | 17.7 | 5.9 (2.4, 9.5) | 1.5 (1.2, 1.9) |
| < = 88% | 361 | 111 | 30.7 | 30.7 | 19.0 (14.1, 23.9) | 2.6 (2.2, 3.1) |
| **BMI** | | | | | | |
| <18.5 | 100 | 23 | 23.0 | 23.0 | 5.3 (-3.5, 14.0) | 1.3 (0.9, 1.9) |
| 18.5–24.9 | 631 | 112 | 17.7 | 17.7 | 0 (ref) | 1 (ref) |
| 25–29.9 | 827 | 115 | 13.9 | 13.9 | -3.8 (-7.7, 0) | 0.8 (0.6, 1.0) |
| 30–39.9 | 1150 | 137 | 11.9 | 11.9 | -5.8 (-9.4, -2.3) | 0.7 (0.5, 0.8) |
| 40+ | 441 | 47 | 10.7 | 10.7 | -7.1 (-11.2, -3.0) | 0.6 (0.4, 0.8) |
| **Lymphocyte, (10^3/ul) (732–8, 731–0, 26474–7)** | | | | | | |
| >1.2 | 1100 | 94 | 8.5 | 8.5 | 0 (ref) | 1 (ref) |
| >0.8–1.2 | 989 | 120 | 12.1 | 12.1 | 3.6 (1.0, 6.2) | 1.4 (1.1, 1.8) |
| 0.5–0.8 | 1113 | 205 | 18.4 | 18.4 | 9.9 (7.1, 12.7) | 2.2 (1.7, 2.7) |
| <0.5 | 398 | 124 | 31.2 | 31.2 | 22.6 (17.8, 27.5) | 3.6 (2.9, 4.6) |
| **Aspartate aminotransferase, U/L (1920–8)** | | | | | | |
| < = 33 | 1530 | 157 | 10.3 | 10.3 | 0 (ref) | 1 (ref) |
| >33 | 2135 | 398 | 18.6 | 18.6 | 8.4 (6.1, 10.6) | 1.8 (1.5, 2.2) |
| **Alanine aminotransferase, U/L (1742–6)** | | | | | | |
| < = 34 | 2318 | 341 | 14.7 | 14.7 | 0 (ref) | 1 (ref) |
| >34 | 1349 | 213 | 15.8 | 15.8 | 1.1 (-1.3, 3.5) | 1.1 (0.9, 1.3) |
| **CRP_cat** | | | | | | |
| 0–1.1 | 2313 | 217 | 9.4 | 9.4 | 0 (ref) | 1 (ref) |
| > 1.1–2 | 912 | 169 | 18.5 | 18.5 | 9.2 (6.4, 11.9) | 2.0 (1.6, 2.4) |
| >2 | 575 | 175 | 30.4 | 30.4 | 21.1 (17.1, 25.0) | 3.2 (2.7, 3.9) |
| **Troponin, ng/mL (10839–9, 42757–5)** | | | | | | |
| <0.1 | 1681 | 229 | 13.6 | 13.6 | 0 (ref) | 1 (ref) |
| 0.1–1 | 507 | 141 | 27.8 | 27.8 | 14.2 (10.0, 18.4) | 2.0 (1.7, 2.5) |
| >1 | 60 | 26 | 43.3 | 43.3 | 29.7 (17.1, 42.4) | 3.2 (2.3, 4.4) |
| **Procalcitonin, ng/ml (75241–0, null)** | | | | | | |
| 0–0.5 ng/ml | 1267 | 163 | 12.9 | 12.9 | 0 (ref) | 1 (ref) |
| >0.5 | 352 | 105 | 29.8 | 29.8 | 17.0 (11.8, 22.1) | 2.3 (1.9, 2.9) |
| **D-Dimer, ng/mL (48066–5 *UNC only)** | | | | | | |
| < = 500 ng/ml | 265 | 22 | 8.3 | 8.3 | 0 (ref) | 1 (ref) |
| 501–1000 | 111 | 25 | 22.5 | 22.5 | 14.2 (5.8, 22.7) | 2.7 (1.6, 4.6) |
| >1000 | 138 | 46 | 33.3 | 33.3 | 25.0 (16.5, 33.6) | 4.0 (2.5, 6.4) |
| **Ferritin, ng/mL (2276–4)** | | | | | | |
| 0–250 | 713 | 58 | 8.1 | 8.1 | 0 (ref) | 1 (ref) |
| >250–500 | 567 | 95 | 16.8 | 16.8 | 8.6 (5.0, 12.3) | 2.1 (1.5, 2.8) |
| >500–1000 | 545 | 103 | 18.9 | 18.9 | 10.8 (6.9, 14.6) | 2.3 (1.7, 3.1) |
| >1000–2500 | 382 | 81 | 21.2 | 21.2 | 13.1 (8.5, 17.6) | 2.6 (1.9, 3.6) |
| >2500 | 122 | 31 | 25.4 | 25.4 | 17.3 (9.3, 25.3) | 3.1 (2.1, 4.6) |
| **C-reactive protein, mg/L (1988–5, 30522–7)** | | | | | | |

*(Continued)*

**Table 4.** (Continued)

|  | N | N death/hospice | stratum % | bar graph of stratum % | % point difference | RR |
|---|---|---|---|---|---|---|
| 0–15 | 411 | 35 | 8.5 | 8.5 | 0 (ref) | 1 (ref) |
| >15–100 | 949 | 114 | 12.0 | 12 | 3.5 (0.1, 6.9) | 1.4 (1.0, 2.0) |
| >100–200 | 533 | 101 | 18.9 | 18.9 | 10.4 (6.2, 14.7) | 2.2 (1.5, 3.2) |
| >200 | 334 | 99 | 29.6 | 29.6 | 21.1 (15.5, 26.7) | 3.5 (2.4, 5.0) |

BMI = body mass index

comorbidities, hypoxia, and level of inflammation, rural patients with COVID-19 were more likely than urban patients to die or to be discharged to hospice. Although our analysis did not elucidate causal reasons for this difference in mortality, the rural health outcome disparities observed are consistent with the literature describing increasing rural-urban disparities in other states and regions [37]. Several studies reported higher COVID-19 infection rates, case fatality rates, and mortality in rural US counties [6–8]. Rural Americans encounter well-documented obstacles to health care access [37–40], which contribute to disparities between rural and urban residents in chronic disease risk factors [41, 42], life expectancy [43], COVID-19 testing [19], and health [44, 45].

Our data showed that rural patients had higher rates of all assessed comorbidities than urban patients. Furthermore, patients with comorbidities had higher rates of death or discharge to hospice compared to patients without comorbidities. In particular, patients in this study with acute myocardial infarction or coagulopathy had the highest percentage of death (29%), which was consistent with previous reports [46]. In addition, a higher percentage of patients from rural areas had multiple comorbidities, as indicated by greater Charlson Comorbidity Index (≥3) and Elixhauser scores (≥5). Both a higher percentage of people with comorbidities and a higher number of comorbidities per person in rural areas match previous reports, that compared to urban residents, people living in rural areas, across all racial and ethnic groups, have higher risks of the five leading causes of death: heart disease, cancer, unintentional injury, chronic lower respiratory disease, and stroke [47–50]. Rural Americans are also more likely than urban residents to have factors linked with hypertension, COPD, and diabetes, such as obesity, poor nutrition, smoking, and alcohol consumption [51, 52]. Several studies have linked hypertension, COPD, and diabetes with a more severe COVID-19 course, ventilation, and death [5, 53, 54]. In our study, all assessed comorbidities, except for uncomplicated hypertension and obesity, were associated with an increased risk of death or hospital discharge.

Patients from rural areas were also more likely to enter the hospital with laboratory evidence indicating hyperinflammation, such as elevated C-reactive protein and lymphopenia, compared to patients from urban areas. These findings suggest that rural patients may have had more severe COVID-19 illness and greater immune dysfunction possibly as a result of greater chronic disease burden, upon hospital admission. Other studies have shown that patients with laboratory markers of hyperinflammation disproportionately developed critical illness compared to those without these markers [46, 55–58].

Although in the United States, the COVID-19 pandemic initially impacted coastal urban areas with the greatest population densities [5, 59], the initial instances of community spread in North Carolina occurred in rural counties [20, 60]. The subsequent rural-urban differences in COVID-19 outcomes in NC are consistent with observations from other states in the Southeast United States. A study of hospitalized COVID-19 patients in Southwest Georgia found that rural populations had a higher prevalence of comorbidities than those described in reports

on urban populations [61]. Huang et al [47] found a higher infection rate and mortality rate in rural counties in South Carolina. Importantly, this latter study also demonstrated a correlation between COVID-19 mortality rates and socioeconomic vulnerability in rural counties [47]. Some of the acceleration in infection rates observed in rural counties may be due to institutional settings with increased transmission risk, including meat and poultry processing facilities, and assisted living centers [59].

This study contributes to the existing literature on health disparities in the United States by providing evidence of rural-urban health outcome differences related to COVID-19. Health inequalities for each of these groups with disproportionate disease burdens may be due to differences in socioeconomic status (SES) [62]. People with lower SES are more likely to have higher rates of comorbidities and limited access to health care resources [62–64]. Taken together, the rural-urban differences in SES and healthcare resources, which elsewhere have been called structural urbanism [20], should be viewed as important predictors of COVID-19 outcomes and evidence for policy changes addressing these differences.

## Strengths and limitations

The primary strength of this study was its analysis of hospital-level data. However, these findings should be interpreted with several limitations in mind. Our data only included patients hospitalized with COVID-19 and did not include those who were ill, but not hospitalized or those who died without a hospital admission (e.g., at home or assisted living facility). Therefore, our data did not include the least and most severe COVID-19 cases. Another limitation in our analysis was that our data included cases only up to the end of September 30, 2020, the point when COVID-19 incidence rates began increasing and accelerating at a faster rate in rural regions than previously seen [65]. Thus, it is possible that our results would have demonstrated greater associations between rurality and death from COVID-19 had we included data beyond September 2020 as larger numbers of people from rural locations were succumbing to COVID-19. Additionally, the effects of subsequent genetic variants of the SARS-CoV-2 virus, including the Delta and Omicron variants of 2021 and 2022, on hospitalizations and deaths could not be determined with data from this time period. Differences in outcomes between rural and urban patients resulting from later variants should be investigated in subsequent research. Furthermore, we were unable to analyze the relationship between the COVID-19 outcome and the timeliness of rural and urban patients' hospitalization after the onset of symptoms because symptom onset was not an available variable in our health system data.

## Policy implications

Several policy recommendations to decrease the rural-urban health outcome disparities for this and future pandemics can be drawn from this analysis. First, improved resources and preparedness for rural areas needs to be prioritized with federal and state funding. Remaining rural hospitals need resources and in areas where closures have occurred or no rural hospitals exist, alternative systems for emergency and acute care should be prioritized. Many rural hospitals were unequipped to manage surges in infectious patients [62, 66]. Relatedly, there needs to be efforts to speed transfer of severe cases to urban hospitals. Increased transportation time and perceived difficulty of travel to physician services are prohibitive to seeking healthcare [47, 67]. The cost (perceived and real) of transport for out-of-network or uninsured patients may also play a role [68] and may be addressed by changes in policy to make this transport free to patients in emergency or life-threatening situations [69].

Second, the COVID-19 pandemic has further revealed the disproportionately high prevalence and severity of chronic diseases in rural areas compared to urban areas. Since

comorbidities caused greater vulnerability to severe COVID-19 illness, improvements to education and public health programs and access to free or affordable health insurance, Medicaid or other, to reduce chronic conditions in rural areas are necessary. Policy makers in states such as North Carolina need a path forward to addressing the barriers to Medicaid expansion or an alternative model to support long-term health of their constituents. Other important components of public health programs would include reducing poverty, reducing the barriers to primary care in rural areas, decreasing mistrust of medical professionals and vaccines, and addressing other economic and social determinants of health resulting in rural-urban health disparities. Rural areas and small towns have lower vaccination rates (36% rural vs 46% urban) [70], and many are now hotspots for COVID-19 infection [71], underscoring the need for trust building and efforts to reduce vaccine hesitancy in rural areas. These policy changes are especially important given the economic challenges faced by rural hospitals, a quarter of which were at high risk of closure prior to the COVID-19 pandemic [66, 72, 73]. The US experienced 181 rural hospital closures in the 15 years prior to the pandemic [74]. Continued closure of rural hospitals would further concentrate health care facilities in large cities [62] and exacerbate existing barriers to healthcare access.

## Conclusions

Rural North Carolina residents hospitalized for COVID-19 had a higher probability of dying or being discharged to hospice in this study. This research adds to the evidence of health disparities in the United States revealed by the COVID-19 pandemic: while many studies have shown racial, ethnic, and age-related disparities, this analysis provides evidence for rural-urban disparities as well. Policies to bolster intensive care units and other medical resources for rural healthcare systems, increase access to primary care, and improve education and public health to attenuate comorbidities in rural areas should be put in place to decrease the risk of death due to future pandemics in rural areas.

## Acknowledgments

Joan Colburn for creation of the map and Bill Ross and Kellie Walters for their guidance on clinical data variable identification and Structured Query Language (SQL) query code creation.

## Author Contributions

**Conceptualization:** Sheri Denslow, Jason R. Wingert, Amresh D. Hanchate, Aubri Rote, Daniel Westreich, Kedai Cheng, Amy Joy Lanou, Jacqueline R. Halladay.

**Data curation:** Kedai Cheng.

**Formal analysis:** Sheri Denslow, Amresh D. Hanchate, Jacqueline R. Halladay.

**Funding acquisition:** Amy Joy Lanou, Jacqueline R. Halladay.

**Investigation:** Sheri Denslow, Janis Curtis, Jacqueline R. Halladay.

**Methodology:** Sheri Denslow, Jason R. Wingert, Amresh D. Hanchate, Daniel Westreich, Laura Sexton, Jacqueline R. Halladay.

**Project administration:** Sheri Denslow, Jacqueline R. Halladay.

**Software:** Kedai Cheng.

**Supervision:** Sheri Denslow, Daniel Westreich, Jacqueline R. Halladay.

**Validation:** Sheri Denslow, Jacqueline R. Halladay.

**Visualization:** Sheri Denslow, Jason R. Wingert, Kedai Cheng.

**Writing – original draft:** Sheri Denslow, Jason R. Wingert, Aubri Rote, Jacqueline R. Halladay.

**Writing – review & editing:** Sheri Denslow, Jason R. Wingert, Amresh D. Hanchate, Aubri Rote, Daniel Westreich, Laura Sexton, Janis Curtis, William Schuyler Jones, Amy Joy Lanou, Jacqueline R. Halladay.

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
