## [Decision Letter · Decision Letter 0]

28 Mar 2022

PONE-D-21-29033Rural-urban outcome differences associated with COVID-19 hospitalizations in North CarolinaPLOS ONE

Dear Dr. Wingert,

Thank you for submitting your manuscript to PLOS ONE. After careful consideration, we feel that it has merit but does not fully meet PLOS ONE’s publication criteria as it currently stands. Therefore, we invite you to submit a revised version of the manuscript that addresses the points raised during the review process.

Please revise.

We look forward to receiving your revised manuscript.

Kind regards,

Academic Editor

PLOS ONE

“The grant was awarded to AL. The NC Collaboratory award does not have an award number. The grants title is Title Back-to-College Challenge: Health Ambassadors for a Coordinated Culture of Safety and Wellness on WNC Campuses and Statewide Co-Morbidity Study with funding from 7/1/2020-12/30/2020.”

7. We note that [Figure 1] in your submission contain [map/satellite] images which may be copyrighted. All PLOS content is published under the Creative Commons Attribution License (CC BY 4.0), which means that the manuscript, images, and Supporting Information files will be freely available online, and any third party is permitted to access, download, copy, distribute, and use these materials in any way, even commercially, with proper attribution. For these reasons, we cannot publish previously copyrighted maps or satellite images created using proprietary data, such as Google software (Google Maps, Street View, and Earth). For more information, see our copyright guidelines: http://journals.plos.org/plosone/s/licenses-and-copyright.

Natural Earth (public domain): http://www.naturalearthdata.com/.

Reviewers' comments:

Reviewer's Responses to Questions

**Comments to the Author**

1. Is the manuscript technically sound, and do the data support the conclusions?

Reviewer #1: Partly

Reviewer #2: Yes

2. Has the statistical analysis been performed appropriately and rigorously? 

Reviewer #1: Yes

Reviewer #2: Yes

3. Have the authors made all data underlying the findings in their manuscript fully available?

Reviewer #1: Yes

Reviewer #2: No

4. Is the manuscript presented in an intelligible fashion and written in standard English?

Reviewer #1: Yes

Reviewer #2: Yes

5. Review Comments to the Author

Reviewer #1: This is a useful descriptive study that is well presented and I have no concerns about the analysis. The concern about inequalities between urban and rural patients is important to examine, particularly in the light of disparities in funding the authors describe.

The main finding is that rural patients are sicker on admission than the urban patients, and adjusting for this moves the association with mortality closer to the null and within the range of being explained by unmeasured confounding. This suggests the key drivers are related to pre existing pre admission factors rather than what happens in hospital. The authors acknowledge this but that their study being hospital based is not able to examine these factors in detail.

However, the major concern is whether there is selection bias with less sick rural patients being less likely to be admitted than less sick urban patients. This is possible because the study is based in tertiary care rather than population based. Therefore, it is important to consider if there could have been any selection bias between rural and urban patients regarding who gets admitted to one of these large academic health systems. Is it possible to have missed less sick rural patients who might have been admitted to local smaller rural hospitals that were not part of the big academic health systems? Or for milder rural patients to have a greater tendency to be managed at home? In which case there would be a trend to only the sicker rural patients being admitted or transferred to one of the 3 large academic health systems. Therefore, there would be selection bias towards sicker rural patients compared to urban patients in the study population.

Which patients would not be eligible for admission to one of these health care systems from the rural and urban populations and would therefore be excluded from the study? If there is a disparity in this related to the rural / urban because of the differences in health insurance then again there could be selection bias towards sicker patients from a rural area because less sick rural patients would not be admitted to these hospitals.

Would there be a difference in decision making for escalation to higher levels care such as ICU, as opposed to palliative hospice care, that would be dependent on the different insurance levels?

Reviewer #2: The article is devoted to the analysis of differences in the outcome of the disease in patients with Сovid-19 from rural and urban areas in in North Carolina. As a result of the study, the conclusion that rural North Carolina residents hospitalized for COVID-19 had a higher probability of dying, which is associated with higher rates of comorbidities in rural patients.

I think that the manuscript can be accepted for publication after some changes have been made:

1. It is necessary to analyze the relationship between the COVID-19 outcome and the timeliness of rural and urban patients hospitalization after the onset of symptoms (patients were hospitalized within a day after the disease, or on the third, fifth, tenth day). It is possible that untimely hospitalization of rural patients is associated with fatal outcomes of Covid-19.

2. It is necessary to add in the Discussion whether the conclusions obtained as a result of the work will change if we analyze the data on patients in 2021-2022. During this period (2021-2022), different genetic variants of SARS-CoV-2 virus circulated in the World, affecting the epidemiological and clinical features of the COVID-19. For example, the Delta and Omicron variants infected younger people that other variants. The Delta variant caused a more severe course of the disease that other variants.

Kind regards.

6. PLOS authors have the option to publish the peer review history of their article (what does this mean?). If published, this will include your full peer review and any attached files.

Reviewer #1: No

Reviewer #2: No

---

## [Author Response · Author response to Decision Letter 0]

3 May 2022

May 2, 2022

Dear Robert Jeenchen Chen, MD, MPH, Academic Editor, PLOS ONE

Thank you for the opportunity to revise and resubmit our original research article (PONE-D-21-29033) entitled, Rural-urban outcome differences associated with COVID-19 hospitalizations in North Carolina. We appreciate the editor’s and reviewers’ valuable input. Our responses (in blue text) to the each of the reviewers’ points is below the reviewer’s statement (indented, black text). All suggested changes have been made in the revised manuscript (highlighted in yellow).

Please let me know if any further information is needed.

Sincerely,

Jason Wingert, PhD, MPT

Professor of Health and Wellness Promotion

University of North Carolina Asheville

We have ensured that our manuscript meets these requirements.

Upon resubmission, we will ensure that all information provided in the ‘Funding Information’ and ‘Financial Disclosure’ sections do match.

The above statement now appears in our cover letter, as requested.

A letter detailing our legal restrictions on publicly sharing these data will be sent to the editor upon resubmission of our revised manuscript. 

A letter detailing our legal restrictions on publicly sharing these data will be sent to the editor upon resubmission of our revised manuscript. These restrictions will also be described in our cover letter, following the prompt 5a above.

The Methods section contains the full names of all reviewing IRBs and a statement clarifying that consent was waived for this study on Page 5-6, Line 147-152.

7. We note that [Figure 1] in your submission contain [map/satellite] images which may be copyrighted. All PLOS content is published under the Creative Commons Attribution License (CC BY 4.0), which means that the manuscript, images, and Supporting Information files will be freely available online, and any third party is permitted to access, download, copy, distribute, and use these materials in any way, even commercially, with proper attribution. For these reasons, we cannot publish previously copyrighted maps or satellite images created using proprietary data, such as Google software (Google Maps, Street View, and Earth). For more information, see our copyright guidelines: http://journals.plos.org/plosone/s/licenses-and-copyright.

Upon submission of the revised manuscript, we will present written permission from the copyright holder to publish these figures specifically under the CC BY 4.0 license, uploaded as an “Other” file with our submission.

We have added this caption to Figure 1.

Reviewers' comments:

Reviewer's Responses to Questions

Comments to the Author

1. Is the manuscript technically sound, and do the data support the conclusions?

Reviewer #1: Partly

Reviewer #2: Yes

2. Has the statistical analysis been performed appropriately and rigorously?

Reviewer #1: Yes

Reviewer #2: Yes

3. Have the authors made all data underlying the findings in their manuscript fully available?

Reviewer #1: Yes

Reviewer #2: No

4. Is the manuscript presented in an intelligible fashion and written in standard English?

Reviewer #1: Yes

Reviewer #2: Yes

5. Review Comments to the Author

Reviewer #1: This is a useful descriptive study that is well presented and I have no concerns about the analysis. The concern about inequalities between urban and rural patients is important to examine, particularly in the light of disparities in funding the authors describe.

The main finding is that rural patients are sicker on admission than the urban patients, and adjusting for this moves the association with mortality closer to the null and within the range of being explained by unmeasured confounding. This suggests the key drivers are related to pre existing pre admission factors rather than what happens in hospital. The authors acknowledge this but that their study being hospital based is not able to examine these factors in detail.

However, the major concern is whether there is selection bias with less sick rural patients being less likely to be admitted than less sick urban patients. This is possible because the study is based in tertiary care rather than population based. Therefore, it is important to consider if there could have been any selection bias between rural and urban patients regarding who gets admitted to one of these large academic health systems. Is it possible to have missed less sick rural patients who might have been admitted to local smaller rural hospitals that were not part of the big academic health systems? Or for milder rural patients to have a greater tendency to be managed at home? In which case there would be a trend to only the sicker rural patients being admitted or transferred to one of the 3 large academic health systems. Therefore, there would be selection bias towards sicker rural patients compared to urban patients in the study population.

We appreciate this comment and our team had similar discussions throughout. However, using limited data sets that were provisioned from existing clinical data warehouses, we cannot provide additional variables that may help to disentangle these many factors that are likely confounders in our analysis. We acknowledge this in our initial manuscript’s limitations section text. 

Which patients would not be eligible for admission to one of these health care systems from the rural and urban populations and would therefore be excluded from the study? If there is a disparity in this related to the rural / urban because of the differences in health insurance then again there could be selection bias towards sicker patients from a rural area because less sick rural patients would not be admitted to these hospitals.

Would there be a difference in decision making for escalation to higher levels care such as ICU, as opposed to palliative hospice care, that would be dependent on the different insurance levels?

Again, with limited data sets from existing clinical data, such important questions cannot be answered, but indeed a mixed methods approach could shed light on these critical questions. 

Reviewer #2: The article is devoted to the analysis of differences in the outcome of the disease in patients with Сovid-19 from rural and urban areas in in North Carolina. As a result of the study, the conclusion that rural North Carolina residents hospitalized for COVID-19 had a higher probability of dying, which is associated with higher rates of comorbidities in rural patients.

I think that the manuscript can be accepted for publication after some changes have been made:

1. It is necessary to analyze the relationship between the COVID-19 outcome and the timeliness of rural and urban patients hospitalization after the onset of symptoms (patients were hospitalized within a day after the disease, or on the third, fifth, tenth day). It is possible that untimely hospitalization of rural patients is associated with fatal outcomes of Covid-19.

We fully appreciate this comment and agree that it would be ideal to understand the time to event from symptom onset to hospitalization, however, symptom onset is not a variable that is available as a discreet field in our health system data, thus we are not able to provide this data. We have included a statement describing this in the limitations section (Page 18, lines 358-361).

2. It is necessary to add in the Discussion whether the conclusions obtained as a result of the work will change if we analyze the data on patients in 2021-2022. During this period (2021-2022), different genetic variants of SARS-CoV-2 virus circulated in the World, affecting the epidemiological and clinical features of the COVID-19. For example, the Delta and Omicron variants infected younger people that other variants. The Delta variant caused a more severe course of the disease that other variants.

Response to reviewer comment #2:

We appreciate this suggestion and indeed the variants have changed since our data collection completed. We have added the following sentence to the Discussion section on Page 18, line 355-358:

Additionally, the effects of subsequent genetic variants of the SARS-CoV-2 virus, including the Delta and Omicron variants of 2021 and 2022, on hospitalizations and deaths could not be determined with data from this time period. Differences in outcomes between rural and urban patients resulting from later variants should be investigated in subsequent research.

6. PLOS authors have the option to publish the peer review history of their article (what does this mean?). If published, this will include your full peer review and any attached files.

Do you want your identity to be public for this peer review? For information about this choice, including consent withdrawal, please see our Privacy Policy.

Reviewer #1: No

Reviewer #2: No

 Additional Reviewer:

Anna Volynkina,

Head of Laboratory of Viral Infections

Diagnostic,

Stavropol Research Antiplague Institute,

355035, Stavropol, Russian Federation

27.03.2022

Dear Dr. Robert Jeenchen Chen,

The paper by Sheri Denslow et al. is devoted to the analysis of differences in the outcome of the disease in patients with Сovid-19 from rural and urban areas in in North Carolina. As a result of the study, the conclusion that rural North Carolina residents hospitalized for COVID-19 had a higher probability of dying, which is associated with higher rates of comorbidities in rural patients. 

I think that the manuscript can be accepted for publication after some changes have been made:

1. It is necessary to analyze the relationship between the COVID-19 outcome and the timeliness of rural and urban patients hospitalization after the onset of symptoms (patients were hospitalized within a day after the disease, or on the third, fifth, tenth day). It is possible that untimely hospitalization of rural patients is associated with fatal outcomes of Covid-19. 

We fully appreciate this comment and agree that it would be ideal to understand the time to event from symptom onset to hospitalization, however, symptom onset is not a variable that is available as a discreet field in our health system data, thus we are not able to provide this data. We have included a statement describing this limitation on Page 18, line 358-361.

2. It is necessary to add in the Discussion whether the conclusions obtained as a result of the work will change if we analyze the data on patients in 2021-2022. During this period (2021-2022), different genetic variants of SARS-CoV-2 virus circulated in the World, affecting the epidemiological and clinical features of the COVID-19. For example, the Delta and Omicron variants infected younger people that other variants. The Delta variant caused a more severe course of the disease that other variants.

Response to reviewer comment #2:

We appreciate this suggestion and indeed the variants have changed since our data collection completed. We have added a sentence describing this limitation to our analysis on Page 18, line 355-358.

Kind regards,

Anna Volynkina

---

## [Decision Letter · Decision Letter 1]

14 Jun 2022

PONE-D-21-29033R1Rural-urban outcome differences associated with COVID-19 hospitalizations in North CarolinaPLOS ONE

Dear Dr. Wingert,

Thank you for submitting your manuscript to PLOS ONE. After careful consideration, we feel that it has merit but does not fully meet PLOS ONE’s publication criteria as it currently stands. Therefore, we invite you to submit a revised version of the manuscript that addresses the points raised during the review process.

Please revise.

We look forward to receiving your revised manuscript.

Kind regards,

Academic Editor

PLOS ONE

Journal Requirements:

Reviewers' comments:

Reviewer's Responses to Questions

**Comments to the Author**

1. If the authors have adequately addressed your comments raised in a previous round of review and you feel that this manuscript is now acceptable for publication, you may indicate that here to bypass the “Comments to the Author” section, enter your conflict of interest statement in the “Confidential to Editor” section, and submit your "Accept" recommendation.

Reviewer #3: All comments have been addressed

Reviewer #4: All comments have been addressed

2. Is the manuscript technically sound, and do the data support the conclusions?

Reviewer #3: Yes

Reviewer #4: Yes

3. Has the statistical analysis been performed appropriately and rigorously? 

Reviewer #3: Yes

Reviewer #4: Yes

4. Have the authors made all data underlying the findings in their manuscript fully available?

Reviewer #3: Yes

Reviewer #4: Yes

5. Is the manuscript presented in an intelligible fashion and written in standard English?

Reviewer #3: Yes

Reviewer #4: Yes

6. Review Comments to the Author

Reviewer #3: This revised article, entitled “Rural-urban outcome differences associated with COVID-19 hospitalizations in North Carolina”, analyzed the differences in the

outcome of the disease in patients with Сovid-19 from rural and urban areas in in North Carolina. And the authors concluded that rural North Carolina residents

hospitalized for COVID-19 had a higher probability of mortality/hospice discharge, after adjustment of age, sex, race/ethnicity, payer, disease comorbidities, presenting oxygen levels and cytokine levels.

I’ve read both original and revised editions, and the authors made a very persuasive explanations and adjustment according to the previous reviewers’ comments. It will be better if it’s added a p value in tables, and clarify the relative ratio using crude model or adjusted model in figures.

Reviewer #4: The article reads well and the revisions suggested have been addressed by the Author/Authors of the Manuscript.

7. PLOS authors have the option to publish the peer review history of their article (what does this mean?). If published, this will include your full peer review and any attached files.

Reviewer #3: **Yes: **Shu-Hsing Cheng

Reviewer #4: No

---

## [Author Response · Author response to Decision Letter 1]

26 Jun 2022

Dear Robert Jeenchen Chen, MD, MPH, Academic Editor, PLOS ONE

Thank you for the review of our original research article (PONE-D-21-29033) entitled, Rural-urban outcome differences associated with COVID-19 hospitalizations in North Carolina. We appreciate the editor’s and reviewers’ continued input. Our response (in blue text) to the reviewer’s additional comment is below the reviewer’s statement (black text). All manuscript revisions are shown in the revised manuscript (highlighted in yellow).

Please let me know if further information is needed.

Sincerely,

Jason Wingert, PhD

University of North Carolina Asheville

Reviewers' comments:

Reviewer #3: This revised article, entitled “Rural-urban outcome differences associated with COVID-19 hospitalizations in North Carolina”, analyzed the differences in the

outcome of the disease in patients with Сovid-19 from rural and urban areas in in North Carolina. And the authors concluded that rural North Carolina residents

hospitalized for COVID-19 had a higher probability of mortality/hospice discharge, after adjustment of age, sex, race/ethnicity, payer, disease comorbidities, presenting oxygen levels and cytokine levels.

I’ve read both original and revised editions, and the authors made a very persuasive explanations and adjustment according to the previous reviewers’ comments. It will be better if it’s added a p value in tables, and clarify the relative ratio using crude model or adjusted model in figures.

Thank you for pointing out that we should specify that Figure 2 is showing unadjusted values. We have modified the Figure heading to state that this is reporting an unadjusted incidence.

For this primarily descriptive analysis, we chose to emphasize estimation over testing through reporting point estimates and confidence intervals and not reporting p-values. Please see articles by Amrhein, Greenland and McShane (Scientists rise up against statistical significance. Nature, 2019) and Ranstam (Why the p-value culture is bad and confidence intervals a better alternative. Osteoarthritis and Cartilage, 2012) for additional support of our decision.

---

## [Decision Letter · Decision Letter 2]

7 Jul 2022

Rural-urban outcome differences associated with COVID-19 hospitalizations in North Carolina

PONE-D-21-29033R2

Dear Dr. Wingert,

We’re pleased to inform you that your manuscript has been judged scientifically suitable for publication and will be formally accepted for publication once it meets all outstanding technical requirements.

Kind regards,

Academic Editor

PLOS ONE

Additional Editor Comments (optional):

Reviewers' comments:

Reviewer's Responses to Questions

**Comments to the Author**

1. If the authors have adequately addressed your comments raised in a previous round of review and you feel that this manuscript is now acceptable for publication, you may indicate that here to bypass the “Comments to the Author” section, enter your conflict of interest statement in the “Confidential to Editor” section, and submit your "Accept" recommendation.

Reviewer #3: All comments have been addressed

Reviewer #4: All comments have been addressed

2. Is the manuscript technically sound, and do the data support the conclusions?

Reviewer #3: Yes

Reviewer #4: Yes

3. Has the statistical analysis been performed appropriately and rigorously? 

Reviewer #3: Yes

Reviewer #4: Yes

4. Have the authors made all data underlying the findings in their manuscript fully available?

Reviewer #3: Yes

Reviewer #4: Yes

5. Is the manuscript presented in an intelligible fashion and written in standard English?

Reviewer #3: Yes

Reviewer #4: Yes

6. Review Comments to the Author

Reviewer #3: This revised article, entitled “Rural-urban outcome differences associated with COVID-19 hospitalizations in North Carolina”, analyzed the differences in the

outcome of the disease in patients with Сovid-19 from rural and urban areas in in North Carolina. And the authors concluded that rural North Carolina residents

hospitalized for COVID-19 had a higher probability of mortality/hospice discharge, after adjustment of age, sex, race/ethnicity, payer, disease comorbidities, presenting oxygen levels and cytokine levels.

I’ve read both original and revised editions, and the authors made a very persuasive explanations and adjustment according to the previous reviewers’ comments. As a result, I have considered this paper ready to be published.

Reviewer #4: The feedback provided for the article titled, "Rural-urban outcome differences associated with COVID-19 hospitalizations in North Carolina" has been addressed and the article can be accepted for publication.

7. PLOS authors have the option to publish the peer review history of their article (what does this mean?). If published, this will include your full peer review and any attached files.

Reviewer #3: **Yes: **Shu-Hsinf Cheng

Reviewer #4: No

---

## [Editor Report · Acceptance letter]

22 Jul 2022

PONE-D-21-29033R2 

Rural-urban outcome differences associated with COVID-19 hospitalizations in North Carolina 

Dear Dr. Wingert:

I'm pleased to inform you that your manuscript has been deemed suitable for publication in PLOS ONE. Congratulations! Your manuscript is now with our production department. 

Kind regards, 

on behalf of

Dr. Robert Jeenchen Chen 

Academic Editor

PLOS ONE